# Digital Devices Use and Sleep in Adolescents: An Umbrella Review

**DOI:** 10.3390/ijerph22101517

**Published:** 2025-10-02

**Authors:** Maria Fiore, Desiree Arena, Valentina Crisafi, Vittorio Grieco, Marco Palella, Chiara Timperanza, Antonio Conti, Giuseppe Cuffari, Margherita Ferrante

**Affiliations:** 1Department of Medical, Surgical Sciences and Advanced Technologies “GF Ingrassia”, University of Catania, 95123 Catania, Italy; desyelucact@gmail.com (D.A.); crisafi.valentina@gmail.com (V.C.); 2School of Specialization in Hygiene, Department of Medical, Surgical Sciences and Advanced Technologies “GF Ingrassia”, University of Catania, 95123 Catania, Italy; vittorio.grieco@pm.me (V.G.); markopa92@hotmail.it (M.P.); chiaratimperanza@outlook.it (C.T.); marfer@unict.it (M.F.); 3Regional Agency for Environmental Protection of Sicily, 90149 Palermo, Italy; aconti@arpa.sicilia.it (A.C.); gcuffari@arpa.sicilia.it (G.C.)

**Keywords:** digital devices, sleep, adolescent

## Abstract

This umbrella review provides a comprehensive synthesis of the available evidence on the relationship between digital device use and adolescent sleep. It summarizes results from systematic reviews and meta-analyses, presenting the magnitude and direction of observed associations. A total of seven systematic reviews, including five qualitative reviews and two meta-analyses, were included, comprising 127 primary studies with a combined sample of 867,003 participants. The findings suggest a negative impact of digital device use on various sleep parameters, including sleep duration, bedtime procrastination, and sleep quality. Devices such as smartphones and computers were found to have a greater adverse effect, while television use showed a weaker association. The most significant disruptions were observed in relation to social media and internet use, with problematic usage leading to delayed bedtimes, shorter sleep duration, and increased sleep onset latency. The review also highlights the role of timing and duration of device use, with late-night use particularly contributing to sleep disturbances. Biological, psychological, and social mechanisms are proposed as potential pathways underlying these effects. Despite moderate evidence supporting the negative impact of digital media on sleep, there is considerable heterogeneity across studies, and many relied on self-reported data, which may limit the generalizability of the findings. Future research should aim to standardize exposure and outcome measures, incorporate objective data collection methods, and explore causal relationships through longitudinal studies. This umbrella review underscores the importance of developing targeted public health strategies, parental guidance, and clinical awareness to mitigate the potential adverse effects of digital device use on adolescent sleep and mental health.

## 1. Introduction

Exposure to digital devices is an increasingly widespread phenomenon starting from the early years of life [1,2]. It was found that 72% of children and 89% of adolescents in the USA have at least one device in their sleeping environment, and most are used before bedtime [3]. Particularly, adolescents use the smartphone for recreational or communicative purposes and as a form of escape or distraction to browse the internet, social networks, listen to music, or have fun together [4]. This rapid overexposure has raised new questions about potential health effects, including sleep problems [5], along with many others, such as loneliness, low self-esteem, poor social relationships, reduced quality of life, poor academic performance, emotional dysregulation, impulsiveness [6], anxiety, stress, and depression [7,8].

Sleep is essential for optimal health and well-being in all stages of life [9]. It plays a vital role in various functions regarding cardiovascular and immune systems, brain function, hormone balance, mental health, and metabolic processes [10]. Sleep is crucial for brain maturation and behavior regulation during adolescence, a critical period for neural development [11,12]. According to the recommendations of the American Academy of Sleep Medicine, adolescents need 8 to 10 h of sleep at night [13]. However, it has been shown that adolescents do not get enough sleep, with a heavy impact on their school, sport, and relationship activities. This phenomenon seems to be related largely to the use of digital devices, especially in the hours before falling asleep. Moreover, some of the sleep problems typical of adolescence (not getting enough sleep or going to bed very late at night and getting up just as late in the morning, falling asleep, or being very sleepy during the day) can be exacerbated using digital devices, especially if they are used in the hours before falling asleep [14,15,16]. Additionally, both the National Sleep Foundation (NSF) consensus panel on adolescent sleep recommendations and the review by Bauducco et al. provide further context and support for the observed associations between digital media use and sleep outcomes in adolescents [17,18].

The new digital devices can have a more specific impact on total sleep duration, sleep quality, delayed bedtime/sleep onset, and daytime sleepiness/tiredness [19] that needs to be clarified. The hypothesis is that such devices might have a negative impact on sleep through a variety of factors. First, they can negatively affect sleep directly by delaying or interrupting sleep time. Furthermore, the content can be psychologically stimulating, and finally, the light emitted by the devices affects circadian timing and sleep physiology [15,20]. This umbrella review aims to systematically evaluate the quality, potential biases, and validity of the evidence regarding the effects of digital device use on adolescents’ total sleep duration, bedtime procrastination, and sleep quality.

## 2. Materials and Methods

### 2.1. Umbrella Review Methods

We followed the Preferred Reporting Items for Systematic Reviews and Meta-Analysis (PRISMA) statement [21] and a methodological guidance [22] to perform this umbrella review. The study protocol was registered in the prospective register of systematic reviews (PROSPERO; registration number CRD42024570963; available at https://www.crd.york.ac.uk/prospero/display_record.php?ID=CRD42024570963; accessed on 11 September 2025). No amendments to the protocol were made after registration.

### 2.2. Search Strategy

The search was conducted using 4 electronic databases (MEDLINE, SCOPUS, WEB OF SCIENCE, and APA PsycARTICLES) in August 2024 and then repeated in August 2025. The search strategy is presented in the Appendix A. Four authors (AD, CV, GV, and TC) independently screened titles and abstracts retrieved from the databases and identified systematic reviews with and without meta-analysis that met the inclusion criteria by full-text reading. Any disagreement in the literature screening between the 4 authors was resolved by a fifth author (MF) through discussion.

### 2.3. Eligibility Criteria

We included systematic reviews with and without meta-analysis, published in English without time period restriction. The inclusion criteria were systematic reviews or meta-analyses, regardless of the design of their primary studies, that evaluated the association of digital device use (smartphones, video games, social media, streaming, computers, consoles, and digital screens) on the following outcomes: sleep quality (bad sleep, sleep deprivation, sleeplessness, sleep inefficiency, and insomnia), total sleep duration, and bedtime procrastination in adolescents aged from 10 to 19 years. For studies with broader age ranges, we only included those that provided disaggregated data for the 10–19 age group. Studies without separate adolescent-specific results were excluded to avoid introducing bias from older age groups.

The exclusion criteria were studies on children younger than 10 years of age, articles in which data were not disaggregated for the pediatric population, studies that did not provide clear information on the statistical methods used, and studies on adolescents with a medical diagnosis according to the Diagnostic and Statistical Manual of Mental Disorders (DSM-5-TR edition). Studies examining conditions not formally recognized as clinical diagnoses, such as problematic device use, were included.

### 2.4. Data Extraction

Data extraction was performed independently by four authors (AD, CV, GV, and TC), and each extraction was reviewed by a different author. Data extraction includes study characteristics (first author, year of publication, study design, methodology, study setting, and sample size), population characteristics (demographics, relationship with study-recruited subject if data are provided by a third subject), exposure details (type of device, frequency and duration of use, usage before sleeping, attitudes, and practices), outcomes (sleep quality, total sleep duration, and bedtime procrastination), and weighted Pearson’s correlation coefficient as an effect measure. A check was performed afterward by another reviewer. Missing full texts was handled by contacting the study investigators. Finally, results of individual studies were presented in standardized summary tables.

### 2.5. Overlap

As proposed by Pieper et al. [23], we estimated the degree of overlap using the Corrected Covered Area (CCA) method. The calculation formula for the CCA index is as follows:CCA (%) = (N − r)/(rc − r)N = Included publicationsr = Number of index publications (primary studies)c = Number of reviews

The result may be interpreted considering the following overlap degree classification. A result between 0% and 5% corresponds to a “slight overlap”, between 6% and 10% to a “moderate overlap”, between 11% and 15% to a “high overlap”, and >15% is considered a “very high overlap”.

### 2.6. Assessment of Methodological Quality of Included Studies

Two researchers (MP and VG) used the assessment of multiple systematic reviews AMSTAR 2 tool to assess the methodological quality. AMSTAR 2 is a 16-item or domain checklist of which seven items are considered critical [24]. Shortcomings in any of the critical domains could affect the overall validity of a review. The domains considered critical were identified using an asterisk. Seven of them were considered critical and were known as “critical domains”. “Critical domains” are modifiable by the authors according to their point of view relative to the articles that have been searched. Therefore, we replaced critical domain number 7 (Did the review authors provide a list of excluded studies and justify the exclusions?) with number 8 (Did the review authors describe the included studies in adequate detail?), highlighted by an asterisk, therefore considering this question more critical to assess the quality of the studies. If the specific criterion was completely met (yes), 1 point was allocated. An overall score relating to review quality was then calculated using the sum of the individual scores. At the end, we rated the overall confidence as reported in Appendix A.

### 2.7. Evaluation of the Strength of Evidence

The strength of evidence was assessed using the GRADE system, where ‘moderate evidence’ is defined as follows: ‘We are moderately confident in the effect estimate; the true effect is likely to be close to the estimate of the effect, but there is a possibility it is substantially different.’ In this review, ‘considerable heterogeneity’ refers to I^2^ ≥ 75%, following Higgins et al., and ‘problematic usage’ refers to a pattern of digital media use associated with functional impairment or distress, including sleep disruption, and meeting the criteria used in the primary studies (e.g., problematic internet use and gaming disorder) [25], which considers five domains that decrease certainty (risk of bias, inconsistency, indirectness, imprecision, and publication bias) and three that increase it (large magnitude of effect, effect of plausible residual confounding accounted for, and dose–response gradient). This systematic evaluation leads to determining the overall certainty of evidence, ranging from high to very low, and has been performed for every outcome. There is no universally agreed-upon method for grading evidence; however, the GRADE approach is widely used and recommended [26,27]. No sensitivity analyses were planned.

## 3. Results

### 3.1. Study Selection

The initial search identified a total of 1740 articles (PubMed 544, Scopus 542) Web of Science 611, and APA PsyNet 43). After removing duplicates, 861 studies were screened based on their titles and abstracts, and 820 studies were excluded. The remaining 41 full texts were assessed for eligibility, leading to the exclusion of 34 studies for various reasons, including the following: wrong age range (n = 23), inappropriate study design (n = 4), wrong population (n = 3), wrong outcome (n = 1), lack of data on exposure (n = 1), and foreign language (n = 2). At the end of the process, seven studies were selected and included in this umbrella review [28,29,30,31,32,33,34] (Figure 1).

### 3.2. Characteristics and Methodological Quality of the Included Studies

Our umbrella review includes 5 qualitative systematic reviews [29,30,31,32,34] and two systematic reviews with meta-analyses [28,33], published between 2015 and 2025. A total of 127 primary studies, involving 867,003 participants, were included. After examining the overlap of primary studies across the systematic reviews included in our umbrella review, we identified 109 different studies. The Correct Covered Area (CCA) value of 2.8% suggests a low degree of overlap. Most of the primary studies included were observational, especially cross-sectional, followed by cohort studies. They were conducted primarily in Europe, followed by Asia and North America. From a practical standpoint, a CCA value of 2.8% indicates that there was very little overlap in the primary studies included across the different systematic reviews. This means that the same primary study was rarely counted more than once in our synthesis, thereby minimizing redundancy and enhancing the breadth of evidence covered.

The details regarding the included studies were summarized in Table 1, while the instruments utilized to measure digital device exposure and outcomes (total sleep duration, sleep quality, and bedtime procrastination) by the authors of primary studies are presented in Appendix A.

The AMSTAR 2 tool highlighted 4 studies [31,32,33,34] rated as “moderate” quality, two studies [29,30] as “low” quality, and one [28] as “very low” quality.

Similarly, the strength of evidence assessment found “moderate” certainty in all studies except a single study rated as “low” strength [28], reflecting some limitations in the available evidence base.

The detailed assessments are presented in Appendix A.

### 3.3. Effects of Digital Device Use on Adolescent’s Sleep by Device Type and Activity

Table 2 shows the effects of digital device use on adolescents’ sleep by device type, study quality, and strength of evidence. To facilitate interpretation, we created an additional visual summary (Figure 2) presenting the main associations by device type and sleep outcome. A color-coded scheme indicates both the strength and consistency of evidence, allowing for a quicker comparison across devices and outcomes.

One meta-analysis [33] included in this umbrella review investigated the relationship between digital media use and sleep health (sleep quality, total sleep duration, and bedtime procrastination). It reported a correlation of r = −0.33 (k = 3, I^2^ = 80.30%) between smartphone use and sleep health, a weaker correlation for social media use (r = −0.12, k = 5, I^2^ = 85.75%), and a correlation of r = −0.19 (*p* < 0.001, k = 6, I^2^ = 97.75%) for problematic digital media use.

Concerning the correlation between digital media activities, such as smartphone use, simultaneous use of multiple devices, and sleep health, they reported a fair correlation (r = −0.33; number of primary studies included k = 3), with considerable heterogeneity (I^2^ = 80.30%) and no publication bias.

Furthermore, they highlighted a weakly adverse role of social media like Facebook and Twitter on sleep health, leading to worse sleep health (r = −0.12; k = 5) with considerable heterogeneity (I^2^ = 85.75%) and no publication bias. Similarly, digital screen use time was very weakly correlated with worse sleep health (r = −0.06, k = 6). Heterogeneity was considerable (I^2^ = 81.77%), and no publication bias was found. In addition, digital media overuse was associated with insomnia symptoms.

Finally, this meta-analysis focused on the negative effect of dysfunctional digital media use, reporting “moderate” evidence of correlation (r = −0.19, *p* < 0.001, k = 6) with considerable heterogeneity (I^2^ = 97.75) and publication bias.

#### 3.3.1. Total Sleep Duration

Five qualitative systematic reviews [29,30,31,32,34] and two meta-analyses [28,33] examined the relationship between adolescents’ electronic device use and total sleep duration. Below are the results by device type.

##### General Digital Screen

da Silva et al. and Gale et al. [31,34] reported “moderate” evidence of an association between digital screen use at bedtime and reduced sleep duration. Similarly, Lund et al. [30] reported that six out of eight studies included in their review confirmed a negative association of total screen time with overall sleep duration. In contrast, Dibben et al. [32] found “inconsistent” findings.

##### Smartphone

Bartel et al. [28] found “low” evidence of association between smartphone use and total sleep duration, reporting a poor negative correlation (r = −0.104; number of included studies =7).

“Moderate” evidence of association was observed by both Lund et al. and da Silva et al. [30,31].

Lund et al. [30], based on three studies, reported a negative effect of smartphone use on sleep duration. da Silva et al. [31], based on five studies, reported reduced sleep duration accompanied by consequent daytime sleepiness.

Dibben et al. [32] confirmed this evidence especially for nighttime smartphone use and smartphone ownership.

##### Television

Bartel et al. [28] found no evidence of correlation between television watching and sleep duration (r = −0.059; k = 10). Similarly, Lund et al. [30] reported “inconsistent” evidence.

##### Computer

Bartel et al. [28] reported “low” evidence of an association between computer use and sleep duration, reporting a poor negative correlation (r = −0.157; k = 4).

Similarly, all the 5 studies considered by Lund et al. [30] found a negative association.

##### Social Media

Lund et al. reported no evidence of association between social media and total sleep duration. Conversely, “modest” evidence of association was found by da Silva et al. [31].

##### Internet Use

Lund et al. [30] provided “moderate” evidence of a negative association between Internet use and sleep duration. Similarly, Kokka et al. [29] showed that increased time spent on the internet led to a greater reduction in total sleep duration, suggesting a dose-dependent relationship. Additionally, they reported shorter total sleep time duration in adolescents with problematic internet use. In contrast, no evidence of correlation (r = −0.087; k = 5) was found by Bartel et al. [28].

##### Video Games

Bartel et al. [28] showed no evidence of correlation between video gaming and sleep duration (r = −0.059; k = 8). On the other hand, “moderate” evidence of negative association was reported by Lund et al. [30].

#### 3.3.2. Bedtime Procrastination

All the studies included in our umbrella review [28,29,30,31,32,33,34] confirmed the positive association between adolescents’ electronic device use and bedtime procrastination, defined as delayed bedtime and delayed sleep onset.

##### General Digital Screens

Lund et al., da Silva et al., and Gale et al. [30,31,34] reported “moderate” evidence of association between digital screens and bedtime procrastination. Specifically, Lund et al. [30], based on nine out of eleven included studies, showed a positive association between increased digital screen use and problems falling asleep/later sleep onset and delayed bedtime.

##### Smartphone

da Silva et al. [31] observed “moderate” evidence of association between smartphone use and bedtime procrastination. Bartel et al. [28] showed “low” evidence of correlation, reporting an (r = 0.131; k = 3).

Concerning sleep onset latency, Bartel et al. [28] found no correlation (r = 0.039; k = 3).

##### Computer

Bartel et al. [28] obtained “low” evidence of correlation between computer use and bedtime procrastination (r = 0.148; k = 2). In contrast, no evidence of association was reported by Lund et al. [30].

##### Television

Bartel et al. [28] showed no significant correlation between television watching and both bedtime procrastination (r = 0.4; k = 8) and delayed sleep onset (r = 0.1; k = 6). Lund et al. [30] reported no evidence of the association between television and bedtime procrastination.

##### Video Games

Bartel et al. [28] found “low” evidence of correlation between video games and bedtime procrastination (r = 0.12; k = 6), while no correlation was reported for sleep onset (r = 0.031, k = 8). “Moderate” evidence of association was found by Gale et al. [34].

In contrast, inconsistent evidence was observed by Lund et al. [30].

##### Internet Use

Bartel et al. [28] observed “low” evidence of correlation between internet use and delayed bedtime (r = 0.212, k = 4) and no evidence of a relationship with sleep onset (r = 0.08, k = 3).

Similarly, both da Silva et al. and Dibben et al. [31,32] showed a “moderate” positive association between this activity and bedtime procrastination. Additionally, Dibben et al. [32] identified a delayed bedtime in adolescents with problematic internet use.

The same evidence was described by Kokka et al. [29]. They emphasized that not only problematic users but also moderate users exhibited increased bedtime procrastination, suggesting that even moderate internet use may be negatively associated with bedtime procrastination.

#### 3.3.3. Sleep Quality

Five systematic reviews [29,30,31,32,33,34] and one meta-analysis [33] investigated the relationship between adolescents’ technological device use and sleep quality.

##### General Digital Screens

Lund et al. [30] found a “moderate” association between digital screen use and sleep quality. In particular, seven out of nine studies included in their review showed the association of electronic device use and sleep parameters such as restless sleep, number of awakenings, and insomnia symptoms. Similarly, da Silva et al. and Gale et al. [31,34] confirmed the association with digital screen utilization at bedtime. Furthermore, they observed that excessive use of digital screens negatively affected sleep quality, leading to more nighttime awakenings and increased daytime sleepiness.

Dibben et al. [32] provided evidence of an inverse association between digital screen overuse and sleep quality; conversely, their findings on regular digital device use were inconsistent.

##### Smartphone

da Silva et al. [31] reported “moderate” evidence of association between mobile phone use and sleep quality, considering smartphones as the device with the higher negative association. Dibben et al. [32] provided the same evidence, attributable to overuse and telepressure (experiencing pressure to socially engage using a mobile phone). In contrast, no evidence was reported by Lund et al. [30].

##### Television

Lund et al. [30] found no evidence of a relationship between television watching and sleep quality.

##### Computer

No evidence of an association between computer use and sleep quality was observed by Lund et al. [30].

##### Social Media

Lund et al. [30] reported “moderate” evidence of a positive association between social media activity, especially overuse, and poor sleep quality. In contrast, Dibben et al. [32] found inconsistent findings, with only a few studies indicating a negative effect of social media on sleep quality.

##### Internet Use

Kokka et al., da Silva et al., and Gale et al. [29,31,34] obtained “moderate” evidence of association between internet use and sleep quality. In particular, Kokka et al. [29] described a positive association between internet overuse and poorer sleep quality, regardless of the device used, highlighting a dose–response relationship. Problematic internet usage was more strongly correlated with sleep disturbances, such as insomnia symptoms and excessive daytime sleepiness.

##### Video Game

Lund et al. [30] reported inconsistent evidence of an association between video gaming and sleep quality.

Finally, we summarize the findings with the highest consistency and evidence certainty, highlighting as follows: (i) the strong association between problematic digital media use (particularly social media and the internet) and poorer sleep outcomes; (ii) the greater disruptive effect of interactive devices (smartphones and computers) compared to passive ones (television); and (iii) the adverse impact of late-night use, especially after lights out. These patterns were observed across multiple high- and moderate-quality reviews (Table 3).

## 4. Discussion

This is the first umbrella review, to our knowledge, providing a comprehensive overview of the available evidence regarding the relationship between digital device use and sleep in adolescents.

Our results highlight disagreement among the studies included.

We identified differences depending on the type of device and type of activity evaluated. Smartphones and computers were types of devices with greater negative impact, while the impact of television on sleep outcomes was weaker. Among the types of use, social media and the internet showed the strongest associations with increased risk of inadequate sleep duration, delayed bedtime, and poor sleep quality [30,31]. These findings are in line with previous studies, suggesting more interactive digital tools have more disruptive effects on sleep compared to traditional devices. This difference may be attributed to their ability to induce greater arousal than passive media [19,35]. However, we also observed inconsistent evidence of the negative association between electronic media exposure and sleep quality and total sleep duration [32].

Timing of digital media use is an important factor that may influence sleep. For example, the meta-analysis by Pagano et al. [33] reported a correlation of r = −0.33 (k = 3, I^2^ = 80.30%) between smartphone use and sleep health and a weaker correlation for social media use (r = −0.12, k = 5, I^2^ = 85.75%). Problematic digital media use showed a correlation of r = −0.19 (*p* < 0.001, k = 6, I^2^ = 97.75%). Using technological devices after turning off the lights leads to a later bedtime, resulting in a reduction in sleep duration [33]. Furthermore, awakenings or notifications from the smartphone during the night may stimulate adolescents both cognitively and emotionally. Social interactions may make it difficult for them to disconnect from electronic media and fall asleep [19,36].

The amount of time spent on electronic devices was also a variable affecting sleep. Higher digital screen use was associated with an increased difficulty falling asleep, with consequent shorter sleep duration and daytime sleepiness [33].

There is a relevant distinction between regular and problematic use of technology. This preoccupant phenomenon occurs when excessive use of digital media leads to behavior and/or physical problems, suggesting an unhealthy or harmful relationship with them. Problematic digital users had a greater risk of developing sleep problems compared to regular users [32,37,38].

We found moderate evidence of the negative effects of problematic social media and internet use, including problematic online gaming, on sleep patterns. In particular, problematic internet use was also associated with symptoms of insomnia and the use of sleep medication [29]. Several authors focused on the adverse impact of online gaming, reporting the association with a poor quality of sleep [39,40]. The World Health Organization (WHO) recognized problematic gaming online as a mental/behavioral disorder, including Gaming Disorder (GD) in the 11th edition of the International Classification of Diseases (ICD-11). However, there is not yet sufficient evidence to justify the inclusion of internet addiction disorder or smartphone addiction disorder as specific diagnostic categories in non-substance addictions, identified by The Diagnostic and Statistical Manual of Mental Disorders (DSM-5) [41].

Regarding video game use, several studies reported sleep interference, decreased sleep duration, and reduced performance upon awakening [42,43,44]. However, evidence of the negative impact of video gaming on sleep was inconsistent in this umbrella review, with variations depending on the specific sleep outcomes evaluated [28,30]. Differences in findings across studies may be explained by several factors, including heterogeneity in how digital media exposure was defined (e.g., total screen time vs. specific activities), variability in sleep measurement tools (objective vs. self-reported), cultural differences in media habits and sleep patterns, and differences in statistical adjustment for confounders such as caffeine intake, mental health, and socioeconomic status. These methodological and contextual variations likely contributed to inconsistent results.

Several mechanisms through which digital devices may interfere with sleep have been suggested. They include biological, psychological, and social factors as follows: (1) evening exposure to blue light from digital devices may delay and reduce the secretion of melatonin, a hormone produced by our body that regulates the circadian rhythm [20]. (2) The sleep–wake cycle can also be disrupted by increased psychophysiological, emotional, and mental arousal caused by screen-based activities, resulting in delayed sleep onset [45]. (3) The pressure to maintain social interactions and the fear of missing out (FoMO) make it difficult for adolescents to disconnect from social media, delaying the time they go to bed [46,47]. Parental behaviors and monitoring also play a role in shaping children’s digital device usage and consequent sleep issues [48]. There is also a lack of understanding in adolescents of the importance of good sleep habits [49].

It is well known that problematic and excessive digital screen use may produce higher levels of anxiety, stress, and depressive symptoms [50]. Sleep has been recognized as a mediator of the relationship between electronic media use and mental health [32,51]. Sleep deprivation associated with digital device overuse increases the risk of mental disorders [52] and suicidal ideation [53,54].

### Strength and Limitations

This umbrella review summarizes and evaluates the evidence on the association between digital device use and sleep in adolescents using a systematic approach. We focused exclusively on adolescents aged 10 to 19 to avoid possible influences due to age differences. It was conducted following the PROSPERO protocol. The overlapping was assessed to eliminate double counting and avoid bias in the results.

Another strength is the inclusion of systematic reviews and meta-analyses published up until August 2025; therefore, most recent studies were included.

In addition, although we restricted our analysis to adolescents aged 10–19 years to ensure population homogeneity, we recognize that this age range includes developmental sub-stages (early adolescence: 10–12 years; middle-to-late adolescence: 12–19 years) that differ in biological, cognitive, emotional, and social characteristics, as well as in lifestyle habits and digital device use patterns. When available in the included systematic reviews, we reported results stratified by narrower age groups or by sex; however, such stratification was not consistently provided, limiting the possibility of performing more detailed subgroup analyses. Future studies should systematically stratify results by developmental stage, sex, and socio-environmental factors to enhance comparability and applicability of findings.

Additionally, while Table 1, Table 2 and Table 3 and Figure 2 provide a structured synthesis of the correlations and strength of associations between different types of digital device use and sleep outcomes, the results could not be further stratified by narrower age groups, sex, or social context. Such detailed stratification was not consistently available in the included systematic reviews or their primary studies. This limitation restricted our ability to present more granular analyses, as suggested by the reviewer. Future research should ensure that data are reported by developmental stage, sex, and socio-environmental variables to enable more detailed and context-specific evidence synthesis.

However, several limitations have been recognized. First, we excluded studies not published in English. Second, a high degree of heterogeneity was observed among the studies or heterogeneity was not reported. Third, included studies showed a great variability between measurement tools used to assess both digital screen exposure and sleep outcome. The lack of standardized methods for data collection makes it difficult to compare groups across studies. Fourth, the findings are primarily based on self-reported measures. Self-reported data should be interpreted with caution as they may not provide an accurate estimate of the frequency of digital devices use and sleep patterns.

Finally, aligned with recent international recommendations on wearable-based sleep monitoring [55,56,57], these technologies, while not a substitute for gold-standard methodologies, can complement traditional assessments by providing objective, longitudinal sleep measures. The World Sleep Society guidelines advocate differentiating standardized core measures from exploratory metrics and encourage collaboration between manufacturers and researchers to ensure measurement accuracy and comparability [55]. The DEVSleepTech guidelines stress the importance of rigorous protocols for device development, validation, and performance evaluation, particularly for consumer-grade devices [56]. Similarly, the Sleep Research Society review offers practical recommendations for the selection and use of wearables in research, highlighting limitations related to proprietary algorithms, inter-individual variability, and potential biases [57]. Integrating validated wearables into future studies on the impact of digital technology use on adolescent sleep could enhance measurement precision and inform targeted public health strategies.

## 5. Conclusions

This umbrella review provides a comprehensive synthesis of the current evidence on the association between digital device use and adolescent sleep, highlighting both the complexity and variability of findings across studies. While the overall evidence supports a negative association, the degree and nature of this impact are not uniform. Timing of use, duration, and the presence of problematic or excessive use emerged as significant modifiers of sleep outcomes.

Biological, psychological, and social mechanisms may underlie these associations, with implications extending beyond sleep to broader mental health outcomes. However, the heterogeneity of study designs, assessment tools, and reliance on self-reported data limit the comparability and generalizability of results. Future research should aim to standardize exposure and outcome measurements, incorporate objective data collection methods, and explore causal pathways through longitudinal designs.

In interpreting these findings, it is important to consider that the available evidence did not consistently provide stratified results by age subgroup, sex, or socio-environmental context. These factors may significantly influence both digital device use patterns and their impact on sleep outcomes. Consequently, while our synthesis supports a general negative association between digital device use and adolescent sleep, the magnitude and nature of this relationship may vary across different demographic and social groups. We recommend that future research systematically report and analyze results by developmental stage, sex, and socio-environmental context to enable more targeted and effective public health interventions.

Given the pervasive role of digital devices in adolescents’ lives, these findings underscore the need for targeted public health strategies, parental guidance, and clinical awareness to mitigate potential sleep and mental health consequences. From a practical perspective, these findings suggest actionable strategies for parents, educators, and clinicians. Limiting smartphone use after lights out, monitoring for signs of problematic or excessive media use, and encouraging earlier cessation of interactive digital activities may help mitigate sleep disruption. Awareness campaigns should also emphasize the stimulating nature of certain content—such as social media interactions or gaming—and its potential to delay sleep onset and impair sleep quality.

## Figures and Tables

**Figure 1 ijerph-22-01517-f001:**
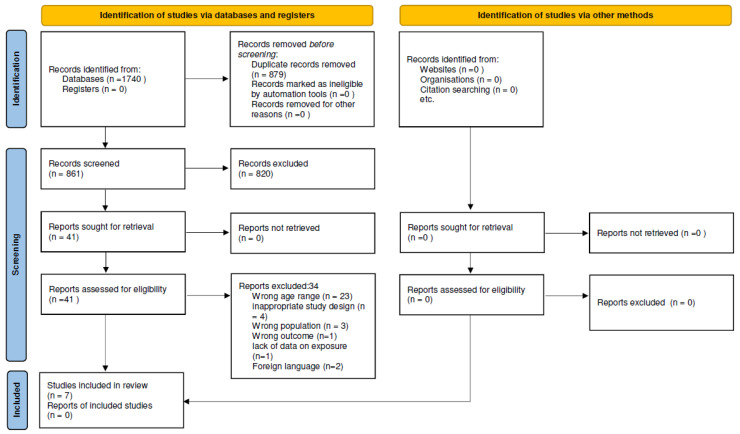
Flow chart of studies identification and selection.

**Figure 2 ijerph-22-01517-f002:**
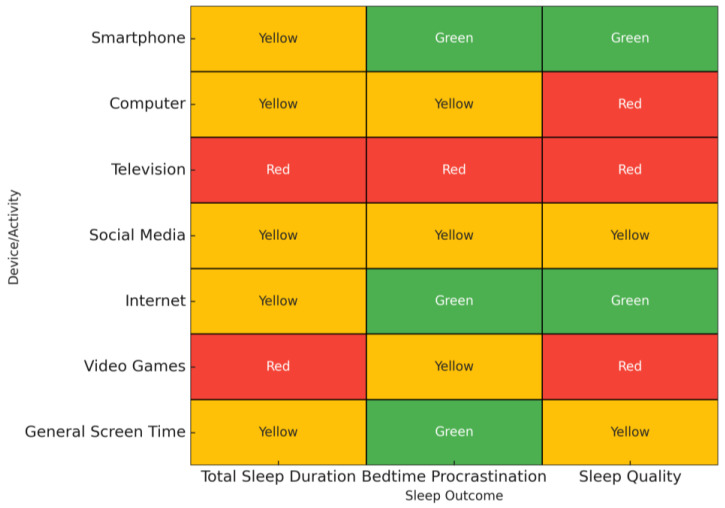
Strength and consistency of evidence by device/activity and sleep outcome. Color-coding: green = strong/consistent evidence; yellow = moderate/partially consistent evidence; red = weak/inconsistent evidence. Data are synthesized from the six included systematic reviews and meta-analyses.

**Table 1 ijerph-22-01517-t001:** Characteristics of included studies.

First Author, Year	Study Methodology	Objective	Database Searched	Publication Date Range	Number and Study Design	Number and Country	Total Participants	Sex	Age (Years)
Bartel, 2015 [28]	Meta-Analysis	Investigating the correlation between video gaming, phone, computer and internet use and evening lighting with late going to sleep	ProQuest Central Flinders University search engine Sage PubMed Google Scholar	2005–2014	17 15 cross-sectional, 2 non-RCT.	USA (4) Japan (3) Sweden (2) Australia (2) England (1) Germany (1) Spain (1) Belgium (1) Israel (1) Taiwan (1)	43,340	F 57.7%	Means from 12.2 to 16.9
Kokka, 2021 [29]	Systematic review	Investigating the impact of problematic internet use on adolescent sleep	PubMed, Scopus	2009–2020	12 12 cross-sectional.	Asia (11) Europe (1)	18,987	F 50.6%	Mean 15.6
Lund, 2021 [30]	Systematic review	Investigating the effects of electronic media use on sleep in children and adolescents	CINAHL, Web of Science, EMBASE, Medline	2009–2019	23, 7 cohort, 14 cross-sectional, 1 non-RCT, 1 RCT.	USA (5), England (2), Australia (2), Finland (2), Netherlands (2), Germany (2), Switzerland (2), Sweden (1), Spain (1), Canada (1), Finland (1), France (1), Denmark (1).	467,202	Not reported	Min 13, max 15
da Silva, 2022 [31]	Systematic review	Investigating the impact of digital screen use on adolescents’ sleep quality	MEDLINE, PubMed, ILACS, SciELO, Scopus, EMBASE, Web of Science, IBECS, Cochrane Library ClinicalTrials.gov, Open Gray	2011–2020	23 3 cohort, 20 cross-sectional.	UK (3), China (2), USA (2), Brazil (2), South Korea (2), Spain (2), Chile (1), France (1), Iran (1), Iceland (1), Japan (1), Lebanon (1), Slovakia (1), Switzerland (1), Turkey (1), New Zealand (1).	153,352	Not reported	Min 10, max 19
Dibben, 2023 [32]	Systematic review	To assess the association between adolescents’ interactive electronic device use with sleep outcomes, and the role of sleep as a mediator or mechanism between screen time and mental health and well-being outcomes	CINAHL, EBSCO, EMBASE, Medline, Web of Science, IBSS, ASSIA	2014–2022	28 16 cohort, 8 cross-sectional, 1 non-RCT, 3 RCT.	Australia (7), USA (6), Korea (3), Switzerland (3), Netherlands (1), Finland (1), Canada (1), Japan (1), Taiwan (1), New Zealand (1), Germany (1).	47,247	F 52.0%	Median of means: 14.8, Min 10, max 19
Pagano, 2023 [33]	Meta-Analysis	Study the relationship between different aspects of digital media use and sleep health pattern	Web of Science, Scopus, PsycINFO, PsycArticles, PubMed, MEDLINE, ERIC, ProQuest, Dissertations and Theses, and GreyNet	2012–2023	23 12 Cohort, 10 cross-sectional, 1 non RCT	USA (7) Netherlands (3) Switzerland (3) Germany (2) China (2) Finland (1) Iran (1) Taiwan (1) 1Kuwait (1) Iceland (1) Korea (1)	116,431	F 53.2%	Mean 13.4, Standard Deviation 1.8 Min 9.6, max 16.5
Gale, 2025 [34]	Systematic review	Examining determinants of poor sleep quality in adolescents	MEDLINE, EMBASE, Ovid, Web of Science, Scopus, PsychInfo, CINAHL, Cochrane Library	2014–2023	7 5 cross-sectional, 1 longitudinal, 1 cohort	UK (2) Canada (1) India (1) Australia (1) Norway (1) Iran (1)	21,205	F 50.9%	Means from 10.0 ± 0.4 to 15.0 ± 1.3

**Table 2 ijerph-22-01517-t002:** Summary of findings for the included studies.

Study, Year	Total Sleep Duration	Quality Assessment (AMSTAR 2) and Strength of Evidence * (GRADE)	Bedtime Procrastination	Quality Assessment (AMSTAR 2) and Strength of Evidence * (GRADE)	Sleep Quality	Quality Assessment (AMSTAR 2) and Strength of Evidence * (GRADE)
Bartel, 2015 [28]	Computer use r = −0.157 (4 **) Internet use r = −0.087 (5) Phone use r = −0.104 (7) Television r = −0.590 (10) Videogames r = −0.059 (8)	Very low confidence Low strength	Computer use r = 0.40 (1) Internet r = 0.080 (3) Phone use r = 0.039 (2) Television r = 0.010 (6) Videogames r = 0.031 (8)	Very low confidence Low strength	Not applicable	Not applicable
Kokka, 2021 [29]	Problematic internet use: negative association (6)	Low confidence Moderate strength	Problematic internet use: positive association (2)	Low confidence Moderate strength	Problematic internet use or excessive internet use: negative association (7)	Low confidence Moderate strength
Lund, 2021 [30]	Computer/internet: negative association (5) Smartphone: negative association (3) Social media: negative association (1), no association (1) Television: negative association (1), positive association (1), no association (2) Total screen time: negative association (6), no association (2) Videogames: negative association (1) no association (2)	Low confidence Moderate strength	Computer/internet: positive association (1), no association (1) Digital screens use: positive association (9) Smartphone: positive association (1), no association (2) Social media: positive association (2) no association (1) Television: no association (2) Total screen time: positive association (5) no association (1) Video games: no association (1), positive (1), and negative association (1)	Low confidence Moderate strength	Computer/internet: no association (2) Smartphone: no association (3) Social media: negative association (3) Television: no association (2) Total screen time: negative association (2), no association (4) Video games: no association (1)	Low confidence Moderate strength
da Silva, 2022 [31]	Computer: negative association (1) Digital screens use: negative association (19) Internet use: negative association (2) Online games: negative association (1) Smartphone: negative association (5) Social media: negative association (1) Television: negative association (4) Total screen time: negative association (3), no association (1) Video games: negative association (3)	Moderate confidence Moderate strength	Digital screens use at bedtime: positive association (1) Internet use: positive association (1) Smartphone: positive association (2) Social media: positive association (2) Total screen time: positive association (3)	Moderate confidence Moderate strength	Digital screens use (10) Internet use: negative association (1) Smartphones: negative association (7) Television: negative association (1) Total screen time: positive association (4) Video games: negative association (1)	Moderate confidence Moderate strength
Dibben, 2023 [32]	Videogames: no association (2), negative association (1) Smartphone: no association (3), negative association (2) Social media use: no association (1) Tecnology use: no association (1)	Moderate confidence Moderate strength	Digital screens evening use: positive association (1) Digital screens use: no association (1), positive association (1) Problematic internet use: positive association (1) Smartphone use at bedtime: positive association (2) Social media: positive association (1) Videogames: No association (1), positive association (5)	Moderate confidence Moderate strength	Computer use: positive association (1) Digital screens use: no association (1) Smartphones: no association (1), positive association (1) Smartphone use at bedtime: positive association (3) Social media: positive association (2) Videogames: no association (1)	Moderate confidence Moderate strength
Pagano, 2023 [33]	Digital screens use at bedtime: weak association (1); Problematic internet use: small association (1) Social media use: no association (1), moderate association (1) Total screen time: no association (2), weak association (1), moderate association (2); Videogames: no association (2), positive association (1)	Moderate confidence Moderate strength	Digital screens use: positive association (1) Social media: no association (2), positive association (1) Total screen time: no association (1), positive association (1)	Moderate confidence Moderate strength	Total screen time: no association (1), small association (1) Social media: no association (1) Problematic use: no association (1)	Moderate confidence Moderate strength
Gale, 2025 [34]	Screen time: negative association (1)	Moderate confidence Moderate strength	Videogames: positive association (1) Screen time: positive associaton (2)	Moderate confidence Moderate strength	Internet use: negative association (1) Screen time: negative association (1) High number of screens: negative association (1)	Moderate confidence Moderate strength

* GRADE (Grading of Recommendations, Assessment, Development, and Evaluations). **High certainty:** We are very confident that the true effect lies close to that of the estimate of the effect. **Moderate certainty:** We are moderately confident in the effect estimate. The true effect is likely to be close to the estimate of the effect, but there is a possibility that it is substantially different. **Low certainty:** Our confidence in the effect estimate is limited. The true effect may be substantially different from the estimate of the effect. r: weighted Pearsons’ correlation coefficient. ** in parentheses: number of studies investigating the risk factor.

**Table 3 ijerph-22-01517-t003:** Summary of the most consistent and high-certainty findings across included reviews, with emphasis on device type, sleep outcome, direction of association, and certainty level based on GRADE.

Device/Activity	Outcome	Direction of Association	Strength/Consistency of Evidence	Notes
Smartphone (general use)	Sleep quality	Negative	High consistency, moderate certainty	Stronger effect with bedtime or late-night use
Smartphone (bedtime use)	Bedtime procrastination	Positive	High consistency, moderate certainty	Linked to later sleep onset and reduced sleep duration
Problematic internet use	Sleep quality and duration	Negative	High consistency, moderate certainty	Dose–response relationship observed
Social media (problematic use)	Sleep quality	Negative	High consistency, moderate certainty	Higher impact with interactive engagement
General screen time (bedtime use)	Bedtime procrastination	Positive	High consistency, moderate certainty	Effect stronger when multiple devices are used
Computer use	Sleep duration	Negative	Moderate consistency, low–moderate certainty	Effect size smaller than smartphones
Television	Sleep outcomes (all)	Mixed/weak	Low consistency, low certainty	Weaker associations than interactive devices

## Data Availability

The data presented in this study are available on request from the corresponding author.

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
