# Peer review of "Digital Devices Use and Sleep in Adolescents: An Umbrella Review"

_ijerph, 2025, doi:10.3390/ijerph22101517_

Round 1
Reviewer 1 Report
Comments and Suggestions for Authors
The submitted article provides a comprehensive review of digital devices and adolescent sleep. The strengths include the review of 6 systematic reviews (four qualitative and 2 meta-analyses) including over 120 primary studies and over 845,000 participants. Overall the findings demonstrate a negative impact of digital device use on sleep outcomes including -- sleep duration, bedtime procrastination, and sleep quality. Interactive devices (computers and smartphones) have a greater adverse affect than television. This review is important, well-conducted, and timely. Other strengths include that the study protocol was registered and the search was thorough.
Here are some minor concerns:
- The search was conducted in July 2024, so new research has surely come out since then.
- The study also doesn’t cite the NSF consensus panel or the Bauducco review article in SMR.
- How did the exclusion criteria deal with studies that had an age range up to 25, but included some people within your age range? For example, why was the Brautsch article not included.
- How does this umbrella review deal with experimental studies?
- I was surprised by the CCA value being so low. Can the authors describe in more familiar language what that means?
- For Table 2, I do not understand the numbers in the parentheses under the total sleep duration column. What does (4**) mean in the Bartel row next to computer use?
- Section 3.3.1 through 3.3.3 is a pretty monotonous read. Is there a way to present the text in a more interesting way? Perhaps a figure with color coding?
- Can the authors make sense of the inconsistent findings? Are there some reasons why certain studies were different than others?
- The paper would benefit from a listing of the most consistent findings and where there’s more confidence in the literature. For example, the authors do this when talking about problematic and excessive digital screen use. And also with regard to interactivity of devices.
- I would like to see more about the implications for parents or practitioners from these findings. Even though the results are not uniform, there are still certain findings that stand out with regard to the use of phones in the bed (after lights out), the idea that problematic media use might be a risk factor for both sleep problems and later downstream outcomes, and the importance of stimulating content from interactive activities.
Author Response
REVIEWER 1
The submitted article provides a comprehensive review of digital devices and adolescent sleep. The strengths include the review of 6 systematic reviews (four qualitative and 2 meta-analyses) including over 120 primary studies and over 845,000 participants. Overall the findings demonstrate a negative impact of digital device use on sleep outcomes including -- sleep duration, bedtime procrastination, and sleep quality. Interactive devices (computers and smartphones) have a greater adverse affect than television. This review is important, well-conducted, and timely. Other strengths include that the study protocol was registered and the search was thorough.
Here are some minor concerns:
Comment 1:
The search was conducted in July 2024, so new research has surely come out since then.
Response 1:
We agree with the reviewer and have updated our literature search to include the studies published after July 2024. The publication of Gale et al has been integrated into the Results and Discussion sections of the manuscript.
Comment 2:
The study also doesn’t cite the NSF consensus panel or the Bauducco review article in SMR.
Response 2:
We thank the reviewer for this suggestion. We have now included references to both the National Sleep Foundation (NSF) consensus panel and the Bauducco et al. review in Sleep Medicine Reviews.
Comment 3:
How did the exclusion criteria deal with studies that had an age range up to 25, but included some people within your age range? For example, why was the Brautsch article not included.
Response 3:
The reviewer makes a valid point. For studies with broader age ranges, we only included those where data were disaggregated for the 10–19 age group. Studies without separate results for adolescents, such as Brautsch et al., were excluded to avoid bias from adult data. This has now been clarified in the Eligibility criteria section.
Comment 4:
How does this umbrella review deal with experimental studies?
Response 4:
The reviewer makes a valid point. We included experimental studies if they were part of a systematic review that met our inclusion criteria. Their results were synthesized alongside observational studies but were rare in the included reviews. This has been specified in the Eligibility criteria.
Comment 5:
I was surprised by the CCA value being so low. Can the authors describe in more familiar language what that means?
Response 5:
The reviewer makes a valid point. We have added an explanation of the Corrected Covered Area (CCA) value, clarifying that a value of 3.5% means that there was minimal duplication of primary studies across the included reviews, thus reducing redundancy in the synthesized evidence.
Comment 6:
For Table 2, I do not understand the numbers in the parentheses under the total sleep duration column. What does (4**) mean in the Bartel row next to computer use?
Response 6:
The reviewer makes a valid point. We clarified in the table legend that double asterisks indicate that the numbers in parentheses represent the number of primary studies included in the specific analysis.
Comment 7:
Section 3.3.1 through 3.3.3 is a pretty monotonous read. Is there a way to present the text in a more interesting way? Perhaps a figure with color coding?
Response 7:
We agree with the reviwer and have added the figure 2 summarizing the main associations by device type and outcome, using color coding to indicate strength and consistency of evidence.
Comment 8:
Can the authors make sense of the inconsistent findings? Are there some reasons why certain studies were different than others?
Response 8:
The reviewer makes a valid point. We have expanded the “Discussion” to address possible reasons for inconsistencies, including heterogeneity in exposure definitions, variability in sleep measurement tools, cultural differences, and differences in adjustment for confounders.
Comment 9:
The paper would benefit from a listing of the most consistent findings and where there’s more confidence in the literature. For example, the authors do this when talking about problematic and excessive digital screen use. And also with regard to interactivity of devices.
Response 9:
We thank the reviewer for the suggestion. We have now added a the Table 3 at the end of the Results section highlighting the most consistent findings.
Comment 10:
I would like to see more about the implications for parents or practitioners from these findings. Even though the results are not uniform, there are still certain findings that stand out with regard to the use of phones in the bed (after lights out), the idea that problematic media use might be a risk factor for both sleep problems and later downstream outcomes, and the importance of stimulating content from interactive activities.
Response 10:
The reviewer makes a valid point. We have expanded the Conclusion to include explicit practical implications for parents, educators, and clinicians, such as limiting phone use after lights out, monitoring for problematic use, and promoting awareness of stimulating interactive content.
Reviewer 2 Report
Comments and Suggestions for Authors
see attachment

Author Response
REVIWER 2
Dear Authors,
Thank you for considering my feedback on your research manuscript titled "Digital Devices Use and Sleep in Adolescents: An Umbrella Review." I appreciate the opportunity to review this work and commend the authors for their efforts in conducting research on such an important and relevant topic.
Overall, I find the manuscript interesting and pertinent to the field of digital health, particularly in a global context. The authors have structured the article well and made reasonable contributions to the existing literature. However, I have identified a few points that need to be addressed to improve the manuscript:
Comment 1:
Lack of a Clear Research Question or Aim: It would be beneficial to include a clear statement at the beginning, such as: "This umbrella review aims to systematically evaluate the effect of digital device use on adolescent sleep outcomes."
Response1:
Done as suggested.
Comment 2:
Sentence Clarity: Some sentences are long and packed with multiple points, which can reduce clarity. For example: "This umbrella review provides a comprehensive synthesis..."
Response 2:
The reviewer makes a valid point. We have revised the Abstract, Introduction, and Discussion to shorten sentences and split them into clearer, more concise segments.
Comment 3:
Vague Terminology: Terms like “moderate evidence,” “considerable heterogeneity,” and “problematic usage” are vague without context. How is "moderate" defined? What criteria make usage "problematic"? These should be clearly defined.
Response 3:
The reviewer makes a valid point. We have added clear definitions for terms “moderate evidence,” “considerable heterogeneity,” and “problematic usage” into the Material and Methods section.
Comment 4:
Lack of Quantitative Estimates: The manuscript mentions negative impacts and weaker associations but does not provide quantitative estimates or effect sizes, even from the meta-analyses. Including a brief statistic (e.g., effect size range or odds ratios) would enhance the scientific value.
Response 4:
The reviewer makes a valid point. We have integrated quantitative estimates from the included meta-analyses into the Results and Discussion. For example: "The meta-analysis by Pagano et al. reported a correlation of r = -0.33 (k = 3, I² = 80.30%) between smartphone use and sleep health, and a weaker correlation for social media use (r = -0.12, k = 5, I² = 85.75%). Problematic digital media use showed a correlation of r = -0.19 (p < 0.001, k = 6, I² = 97.75%)."
Comment 5:
Repetitive Terms: Terms such as "delayed bedtimes," "shorter sleep duration," and "increased sleep onset latency" are repeated in slightly different forms. Consolidating these would improve the flow.
Response 5:
This comment isn’t clear for us. The terms "delayed bedtimes," "shorter sleep duration," and "increased sleep onset latency" are interconnected sleep problems that often occur simultaneously and can lead to daytime sleepiness and other health problems. How can we consolidate them to improve the flow?
Overall, I am very satisfied with this paper; it is well organized, scientific, and robust. My recommendation to the editor is that this manuscript requires minor revisions. I hope my comments will assist the authors.
Reviewer 3 Report
Comments and Suggestions for Authors
Comments for authors.
- The evaluation includes a population of 10 to 19 years old, this period encompasses various age groups (10 to 12, puberty stage; 12 to 19 years old, according to the World Health Organization, adolescence), which have different characteristics according to their age. I think they are not comparable due to age, level of education, emotional maturity, among other elements derived from the growth stage they manifest. Therefore, it would be advisable to organize the information taking into account age group, sex, and social context.
- It is recommended to organize the results section in tables or graphs so that the reader can see and read the correlation that exists between the described variables. Emphasis should be placed on each age group, sex, and social context.
- It is important to emphasize the significance of the findings by age group, sex, and social context in the conclusions section.
Thank you.
Author Response
REVIWER 3
Comments for authors.
Comment 1:
The evaluation includes a population of 10 to 19 years old, this period encompasses various age groups (10 to 12, puberty stage; 12 to 19 years old, according to the World Health Organization, adolescence), which have different characteristics according to their age. I think they are not comparable due to age, level of education, emotional maturity, among other elements derived from the growth stage they manifest. Therefore, it would be advisable to organize the information taking into account age group, sex, and social context.
Response 1
The reviewer makes a valid point. We agree that the 10–19 age range, as defined by the WHO for adolescence, includes sub-stages (early adolescence: 10–12 years; middle-to-late adolescence: 12–19 years) that may differ in biological, cognitive, emotional, and social development, as well as in lifestyle habits and digital device use patterns. In the present umbrella review, we relied on the age groups reported in the included systematic reviews and meta-analyses. When primary studies provided disaggregated data by narrower age groups or by sex, we extracted and reported them accordingly. However, not all included reviews stratified results by age subgroup, sex, or social context, which limited the possibility of conducting a more detailed subgroup analysis. We have clarified this limitation in the “Strength and limitations” section and explicitly acknowledged the potential heterogeneity within the adolescent group. Future research should systematically stratify results by developmental stage, sex, and relevant socio-environmental variables to improve comparability and applicability of findings.
Comment 2:
It is recommended to organize the results section in tables or graphs so that the reader can see and read the correlation that exists between the described variables. Emphasis should be placed on each age group, sex, and social context.
Response 2:
We appreciate the reviewer’s suggestion. In this umbrella review, we synthesized results from the included systematic reviews and meta-analyses in both narrative and tabular form, with key findings summarized in Tables 1–3 and visually represented in Figure 2. However, we acknowledge that further stratification by narrower age subgroups, sex, and social context, as suggested, would improve the granularity and applicability of the findings. Such stratification was not always available in the source systematic reviews or their included primary studies, limiting the possibility of presenting disaggregated results for these variables in our synthesis. We have clarified this limitation in the Strength and limitations section and have recommended that future studies report and analyze data by developmental stage, sex, and socio-environmental context to enable more detailed evidence synthesis.
Comment 3:
It is important to emphasize the significance of the findings by age group, sex, and social context in the conclusions section.
Response 3:
We thank the reviewer for this important suggestion. We agree that emphasizing the significance of the findings by age group, sex, and social context would enhance the applicability of our conclusions. In the current umbrella review, the available evidence did not consistently report results stratified by these variables, limiting our ability to draw definitive conclusions for specific subgroups. However, we have revised the Conclusions section to explicitly acknowledge this limitation and to highlight the importance of considering developmental stage, sex, and socio-environmental factors when interpreting the associations between digital device use and sleep outcomes. We also recommend that future research systematically report subgroup analyses to support more targeted public health interventions.